



# The Measurement of Knowledge Transfer

Thomas von Clarmann[1]

[1]Karlsruhe Institute of Technology, Institute of Meteorology and Climate Research,
Karlsruhe, Germany

**Abstract.** The measurement of knowledge transfer is considered an important component of the overall performance assessment of research groups. It is, however, not a trivial task, because there is agreement on neither the definition nor on the logical structure of knowledge. In this paper the problems related to the explication of the title term are summarized and the relation between knowledge and information is critically discussed. Open questions with respect to the logical structure of knowledge
and its transfer are identified. Requirements to the concept of a knowledge transferometer are developed. Finally the request to scientests to measure their knowledge transfer is critically discussed.

**Keywords.** Knowledge transfer

   Project evaluation

   Outreach

Information

## 1 Introduction

In order to get the largest possible benefit for the public money spent in geo-scentific activities, research ministries of some countries or international funding agencies have installed various levels of science management to control scientific progress. Formalized assessment of scientific progress by evaluation panels is an essential part of this. Evaluation of scientific work
does not only assess the scientific success in a narrower sense but also aims at quantifying the outreach of the respective project. One particular request within the quantification of outreach is the 'measurement of knowledge transfer'. Since this request occasionally leaves the geo-scientist in charge somewhat clueless, this paper aims at clarification of related notions and concepts. It will be tried to apply the same rigour which is usually applied to laboratory or field measurements in geo-sciences.

   In the next section it will be attempted to explicate the notion of 'knowledge'. Section 3 deals with the concept of 'transfer'.
The possibility of related 'measurements' will be critically discussed in Section 4. Finally, in the concluding section, the findings will be summarized, conclusions on the possibility of the measurement of knowledge transfer will be drawn, and important future work will be identified.




## 2   Concepts of Knowledge

There exists a vast literature on knowledge and its growth, e.g., "Criticism and the Growth of Knowledge" by Lakatos and Musgrave (1970) or "Error and the Growth of Experimental Knowledge" by Mayo (1996), just to name a few. Nevertheless, the exact meaning of the term 'knowledge' remains somewhat vague. According to Plato (369BC), knowledge is 'justified true

belief'. Although Gettier (1963) has shown that the Platonian definition does not suffice to exclude beliefs which are, accoding to common sense, not knowledge, they seem at least to define three necessary conditions. Numerous additional conditions were suggested, but the discussion of these is beyond the scope of this paper. It is, however, not quite clear how far these attributes of knowledge refer to knowledge in its general sense or to scientific knowledge. More recent explication of knowledge highlights that there exist multiple variants of knowledge (Blackler, 1995). Knowledge transfer in the context of this study, i.e., knowledge

transfer in the context of the evalution of scientific, and in particular geo-scientific, success, seems to be a transfer of scientific knowledge into general knowledge. Thus the characterization of scientific knowledge in narrow sense might be too limited for the purpose of this study.

In this paper it will be tried to stay as neutral as possible with respect to the semantics of the term 'knowledge' and to concentrate on the logical structure of the related concept instead. The reason is roughly this. In order to define a transfer

function for knowledge, it is not necessarily relevant to explicate the term knowledge semantically at full rigour. To make this understandable, a severe misconception has to be cleared up: In the context of measurement of knowledge transfer we are not actually interested in knowledge itself, i.e., the content of the knowledge, but only in the size or amount of knowledge. Similarly, to establish rules how to count apples, I do neither need to know their DNA nor their content of vitamins. A number of questions with respect to the logical structure of 'knowledge' still remains unanswered by previous work. In the following

it will be tried, without claiming to be exhaustive, to shed some light on some of these open issues.

### 2.1   The Ontology of Knowledge

The first obvious question with respect the question what 'knowledge' actually is must be if this term belongs to the object language or to the meta-language. In other words, it has to be clarified if 'knowledge' as assessed here exists in the real world or if it is part of a meta-theory used to describe the real world. There seem to be multiple valid answers to this question and

a full-blown discussion of this issue requires a hierarchy of meta-languages as suggested by Tarski (1944). The fact that we can have knowledge about the world suggests that 'knowledge' belongs to the meta-language. However, in the context of this work, knowledge and its transfer are the objects of our investigation. In the context of measurement of knowledge transfer it seems thus to be adequate to consider knowledge as a term of the object language. This judgement is a pragmatic choice and is not meant to deny that the term 'knowledge' can be constituted also as a meta-theoretical term.



## 2.2 Knowledge and Information

*Prima facie* there seems to be not much difference between 'knowledge' and 'information.' In the following, however, it will be shown that this notion is untenable. 'Information' is a well defined term (Shannon, 1948)[1]. Roughly speaking, it measures the potential reduction of the uncertainty with respect to some uncertain prior knowledge. Without prior knowledge, the Shannon

information is not a meaningful concept. The reason is roughly this. Zero prior information is equivalent to infinite uncertainty. One bit of Shannon information reduces the uncertainty by a factor of two. Infinity divided by two still is infinity, so nothing is gained. A paradigmatic example is this. There are eight boxes and in one of them there is an item which shall be found. With one correct answer to the question "is the item in one of the boxes #1 to #4?" the uncertainty about where the item is is reduced by a factor of two. This is one bit of information. With three bits of information, the correct box can be identified. However,

without the prior knowledge that the item is in one of the eight boxes, this procedure makes no sense and information is an ill-defined quantity, i.e., zero prior information constitutes a singularity in information theory.

Further, some problems regarding the relation between knowledge and information arise because here the term 'information' is often used equivocally for the accumulated (integrated) and the incremental (differential) quantity. Here, a terminology is used where 'information', in consistence with the above, is used exclusively for an incremental designate, in the sense of

'information gain'. As opposed to that, the term 'knowledge' is used exclusively for an accumulative designate. For a person who can, with unlimited efficiency, absorb information, has the necessary prior knowledge to make use of the information received, and also can reliably distinguish information from des-information, one could say that the received information measures the increase of knowledge. According to this notion one could define knowledge as the integral over information. Putting the question about the extensiveness of knowledge aside for a moment, this still is, however, an idealized case which

often has not much to do with real life. Assume a newspaper in a language the reader does not know: Putting aside problems of the kind that newspapers often publish lies or rumors and assuming instead in a friendly way that this newspaper contains only true information, still attempts to read this newspaper do not increase the knowledge of the reader because she cannot add the information contained in the newspaper to his/her knowledge. Further, if the person refuses to read the foreign newspaper, *a fortiori* the knowledge of the person is not increased by any means. To provide a useful link between information and

knowledge gain it is necessary to make use of a kind of information resorption function which controls which fraction of the available information adds to the knowledge of the knowledge bearer. From this concept it becomes clear that knowledge is a subjective quantity, this is to say, a quantity associated with a person rather than to a medium or a situation.

There remains, however, one further problem: information according to Shannon (1948) is always positive. This theory of information leaves no room for situations where one has, at a given time $t$, less knowledge then at any prior time $t - \Delta t$.

Regarding now the lies which might be distributed by some TV stations, and assuming that the consumer believes them, it is easily possible that the knowledge at a later time is smaller than the knowledge at an earlier time. A quantitative theory of knowledge thus must consider concepts like 'false information', 'alternative facts', 'loss of knowledge' and similar.

---

[1]An alternative approach is the theory of semantic information by Carnap and Bar-Hillel (1952)





Further problems arise when the knowledge bearer does not believe the new information or does not have the necessary prior knowledge to make any sense of the new information provided. All these problems suggest that the concept of 'knowledge' has a stronger link to the real world than 'information' in a sense that 'knowledge' seems to have a closer link to what is true in the real world, while 'information' is a more formal concept, existing more independently from the real world.

5    The situation becomes even more complicated by the fact that knowledge does not only exist on things existing in the real world but also on formal constructs like theories, formal axioms, definitions or mathematical theorems. It has to my best knowledge[2] not yet been shown that this kind of knowledge can in any coherent manner be reduced to the concept of accumulation of information in a Shannon sense. We leave this issue open for future research and just reserve the term 'formal knowledge' for knowledge in this field. It is interesting that formal knowledge can thus be generated and multiplied without any experience from the 'real world', which seems to challenge Locke (1690) and his notion of non-existence of innate ideas. There are at least three possible approaches to solve this conflict: One is to consider all formal knowledge as abstracted structures which (or the components of which) can be reduced to isomorphic structures in the real world. The second is to consider formal knowledge as a kind of attributes of human beings, who in themselves are, with their ideas and theories, etc, part of the world of things. The third is simply to extend the intended applicability of the term 'knowledge' to entities beyond the world of things.

15    A further major difference between the concepts of 'knowledge' and 'information', which has already been insinuated above but not discussed at any depth, is that knowledge is subjective while information is objective. Subjectivity is implied by one of the three Platonian necessary conditions, i.e. that knowledge can only be if it is believed by the knowledge bearer. This further implies that knowledge cannot exist without a knowledge holders. On the contrary, information is an objective quantity, which can be there (e.g. in a book, a newspaper, a hard-disk, or a traffic sign) even if it is not realized, appreciated, or believed by anybody (Ben-Naim, 2008).

After meandering around these various aspects of the essence of 'knowledge' and still neglecting problems in what sense knowledge is an extensive quantity, we summarize:

- resorption of accessible true information increases knowledge.

- information which is not believed does not increase knowledge.

- information which is not understood does not increase knowledge.

- the increase of formal knowledge, in which manner it may be achieved, also contributes to the increase of total knowledge.

- resorption of accessible des-information (i.e. false information) decreases knowledge.

- forgetting decreases knowledge.

- erroneous thinking can decrease the formal knowledge.

---

[2]The author admits to be guilty of circularity because of using one of the title-terms of the investigation also in his meta-language. The implied circularity, however, has no influence on the key results of this paper



This list, without claiming to be exhaustive, summarizes the components of knowledge loss and gain identified so far. As a preliminary working definition, without claiming its final adequacy, we suggest "knowledge is the ability to assign justified truth values to statements." Each assignment of a truth-value is the answer to a binary question.

## 2.3 Is Knowledge an Extensive Quantity?

In a Platonian sense, knowledge certainly is not an extensive quantity, because it is not a quantity at all. Knowledge in the Platonian sense characterizes the content, not the amount. In the context of measurement of knowledge transfer, as said above, not the content but the amount of knowledge is in the focus of the interest.

In the previous subsection, knowledge was tentatively explicated as the integral over the accessible (positive or negative[3]) information. Prior to the clarification of some of the characteristics of the entity 'knowledge', this can only be understood in

a metaphorical rather than verbatim way. One of the central questions is in what sense 'knowledge' is an extensive quantity. With Carnap (1995/1966) we distinguish here between additive and non-additive extensive quantities.

First, the stronger condition, the additive extensiveness is discussed. With $K$ being a knowledge function and $a$ and $b$ being two knowledge holders, then additive extensiveness would require (Carnap, 1995/1966):

$$K(a \circ b) = K(a) + K(b) \tag{1}$$

which says that the knowledge both knowledge bearing persons have as a team is the sum of the knowledges (or whatever the correct plural of 'knowledge' is) of each of the two knowledge holders. Additive extensiveness of knowledge can easily be refuted with the following simple example: Let the knowledge of person $a$ be the knowledge of French language. Let further the knowledge of person $b$ be the words of a French song, learned phonetically without knowing the language. If persons $a$ and $b$ form a team, then they have, without using any further knowledge, a chance to understand the meaning of the words, which

they do not have as separated individuals. Thus we have, for positive $K(a)$ and positive $K(b)$:

$$K(a \circ b) > K(a) + K(b) \tag{2}$$

Another counter-example is combination of overlapping, thus redundant, knowledge. Assume a password containing eight characters. Assume that two persons each know four of these characters. *Prima facie* it seems that the information should be sufficient but what if the first person knows characters one to four, and person two knows characters two to five? These counter-

examples clearly rule out additive extensiveness but leaves open the possibility of any other extensiveness. Considering that it is still under debate what knowledge actually is (Gettier, 1963) and how inadequate definitions of knowledge can be "degettiered", and having only a vague idea how the amount of knowledge can be quantified, it is not easily possible to accurately specify the extensiveness of knowledge (or more precisely: of the amount of knowledge). Given the task of measuring knowledge transfer, it seems justified to assume as a working hypothesis that the amount of knowledge transferred is an extensive (but not

necessarily an additive) quantity.

---

[3]negative information is meant as a pre-scientific term, associated with the decrease of information associated with des-information, forgetting, or erroneous thinking.



## 2.4 The Dimension of Knowledge

While 'information' is always related to a certain question or quantity, 'knowledge' is about everything the knowledge-bearer knows. Despite the inherent circularity in the above statement, it is obvious that knowledge exceeds information not only in a sense that it is the accumulated (i.e. integral) quantity rather than the incremental (i.e.. differential) quantity but also in a sense that it is associated with multiple rather than single questions or quantities. Admittedly the Shannon concept is also applicable to multiple quantities by utilizing vector algebra. The gain of information is the difference of the entropy before the information is received and the entropy after the reception of knowledge. The entropy difference is scalar even for multidimensional states, thus the information is a scalar quantity. It suggests itself to treat knowledge transfer in a similar way. Each such transformation, however, requires a kind of norm or metric, whose definition implies its own problems.

## 2.5 Measurement of Knowledge

*Prima facie* quantification of knowledge could be achieved just by counting the true statements the knowledge holder can make. This approach, however, leads immediately to absurdities, since the adequate weight of the statements this concept is based on cannot be assumed to be equal. Let person 1 know that all monkeys have a backbone. Let person 2 know that all mammals have a backbone. Who knows more? Thus, it is obvious that it is not only the number of true statements that counts but also the extension of each statement. We find that the measurement of knowledge in unities which somehow can be reduced to single problems leads to major practical problems. Thus we put this issue aside and search for a more practical solution.

## 2.6 An Operational Definition of Knowledge?

Within empiricism and positivism (von Mises, 1939) it has been the standard approach to clarify problematic quantities using an operational definition (Bridgman, 1927), i.e., the quantity is defined using a measurement instruction. Later this approach has been heavily criticized, e.g., by Suppe (1974).

In order to fairly characterize the semantic content of knowledge, its multi-dimensionality (i.e. consideration of knowledge on different things) has to be taken into account. The amount of knowledge, however, shall be a scalar quantity. The aspect of multi-dimensionality poses some problems on the operational definition of knowledge, and the metric mentioned in the previous section is an essential part of the definition of the measurement system. We put this issue aside for a moment and turn towards more practical considerations.

Taking into account that knowledge is a characteristic of a person, namely the knowledge-bearer, it suggests itself that knowledge be measured by a quantitative analysis of how the test person fulfills certain tasks, namely specific tests. Crossword puzzles may be considered a candidate for such a test, more sophisticated tests certainly are available. The detailed specification of such tests are beyond the research field of philosophy of science or epistemology; it is considered to be a task for psychology instead. One problem, however, remains: Each test is limited to a certain field of knowledge, and the weights of the different fields of knowledge among each other, as well as the weights between the different qualities of knowledge (reproductive knowledge, mental abilities, skill, workmanship, methodical knowledge) will always remain a subjective choice.





Further, there is the problem that can be called the teleological trap: The test person who wants to perform well in a particular test may prepare herself for this specific test. Thus a test, which was representative in a sense that the tasks were a representative sample of knowledge indicators, will be no more representative, but biased after the test person's preparation. Standards to prevent this have been developed (American Educational Research Association, American Psychological Association & National Council on Measurement in Education , 1999).

Beyond the problem of the adequate content and design of appropriate tests, there is also the problem that some persons underperform in tests, i.e. their abilities are much better than what the result of a test suggests. Psychological tests of knowledge never test the knowledge of a person but instead they test the ability to reproduce or apply knowledge under test conditions. Thus, a knowledge calibration coefficient is needed to reconstruct the knowledge of a person from the result of the psychological knowledge test. This is, of course, still a crude simplification, because the knowledge calibration factor is not only person-dependent but also mood-dependent, health-dependent, depending on the time of the day, and others.

I am not aware of any test which is actually adequate to measure the total knowledge of a person. It seems not even clear how the total scientific knowledge of a person shall be defined. Thus I consider the approach to measure a person's knowledge as failed, at least for the moment.

## 2.7 Knowledge as Theoretical Quantity

Problems identified in the previous sections to reduce knowledge to well-defined and measurable quantities like information suggest that it might be adequate to treat the term 'knowledge' as a 'theoretical term' in the sense of Carnap (1995/1966). Using Ramsey-elimination (Ramsey, 1929), any statement about 'knowledge' is reformulated as an existence statement of a quantity $x$ which replaces the term 'knowledge' in all statements where it appears. According to J. Sneed (1979/1971), the existence of an unambiguous solution for $x$ proves that the theory involving the theoretical term under assessment is an empirical theory. If there exist multiple solutions for the undefined term $x$, the theory is at least fulfillable or self-consistent. If it can be shown that no solution can be found for the undefined quantity $x$, then the theory is inconsistent and thus analytically false. While interesting in its own right, this concept does not help to provide a methodology to measure knowledge.

## 3 Transfer

In pre-theoretical terminology 'transfer' is a process where a transferendum is transferred to a destination of the transfer. A paradigatic example is radiative transfer (Chandrasekhar, 1950). It describes how much radiative energy in a certain frequency interval is received at a certain point in space along a certain line of sight, under consideration of emission, absorption and scattering along the line of sight. In the following it will be tried to apply similar concept to 'knowledge transfer'. Recent literature discusses means to achieve knowledge transfer and the benefit through knowledge transfer but falls short of establishing an adequate and universal quantification of knowledge transfer (Argote and Ingram, 2000; Kane et al., 2005).



## 3.1 Explication of the term

In common language, transfer is a process, not a quantity. Thus, 'knowledge transfer' is a process which involves two persons or groups of person. Person(s) #1 is called the knowledge transmitter. She makes, consciously or unconsciously, her knowledge available to person(s) #2, the knowledge receiver. The process of knowledge transfer includes all actions undertaken to change

the efficiency of knowledge transfer. These include 'positive' actions like didactic preparation of the knowledge, or 'negative' actions like encryption. The amount of knowledge transfer must be a function of the amount of knowledge communicated by the knowledge transmitter and an efficiency function $f$ (because not everything said by person #1 is necessarily understood by person #2) The increase of knowledge of person #2 due to the knowledge transfer process $\alpha$ should then equal the knowledge communicated by person #1 times the efficiency $f_\alpha$. When information is communicated to multiple persons, it may be worth-

while to sum up related gains of knowledge increases by all involved persons. In that sense, the amount of knowledge transfer seems indeed to be an extensive additive quantity.

## 4  Measurement

Given that the measurement of 'knowledge' is still an unresolved problem, the measurement of the derived quantity 'knowledge transfer' obviously is a challenge. There are multiple options.

Option 1 is the measurement of the primary quantities, *viz.*, the knowledge of a person before and after the process of knowledge transfer. From these measurements the knowledge transfer is inferred by simply calculating the difference. This is not as simple as it appears, because first the measurement of knowledge is still an unresolved problem, and second, the number of knowledge receivers per knowledge transmission is not at all clear. When a research article is published, it is not at all clear how many people read it, and even less clear which fraction of these really understands it. Even the determination of the

knowledge content of the article is a challenge in itself. Further, readers who have known all the content before do not learn a lot from the article, thus the complementarity of the knowledge is an issue. Thus this approach can be regarded as non-realistic.

Option 2 is to define knowledge transfer operationally via a measurement instruction. This approach is henceforth referred to as 'direct measurement of knowledge transfer'. The idea is to define a certain reference knowledge transfer process and to define the amount of knowledge transfer on the basis of this reference process. The fact that the measurement might not be

repeated under equal conditions and that the ideal measurement procedure can in reality only be followed approximately adds some non-negligible uncertainty and puts this approach in the vicinity of *ceteris paribus* laws (Schurz, 2002, and references therein). Any setup which is adequate for such a measurement is henceforth called 'knowledge- transferometer'.

## 4.1  The unit of knowledge transfer

Since neither knowledge nor even the amount of knowledge are the same as information, it seems inadequate to use the same

unit for information and knowledge. Thus, I suggest to name the unit of knowledge transfer "Kant". Further I suggest the



following quasi-operational definition: One Kant is the gain of knowledge achieved by an average high school graduate after reeding the "Critique of Pure Reason". The reduction of this to SI units is left to more ambitous knowledge metrologists.

## 4.2 Calibration of the Knowledge Transferometer

The reference knowledge transferometer might not be adequate to measure knowledge transfer in all situations. Knowledge

transferred by publication of a scientific journal article needs most likely to be measured by a different setup than knowledge transferred by a soccer coach when he/she explains his/her strategy to the team. Thus different measurement setups go under the over-arching term knowledge transferometer. In order to make measurements with the various knowledge transferometers comparable, some calibration is needed. In other words, a rule must be defined how to find out how many milliKants or picoKants the amount of knowledge transferred and measured with one of these knowledge transferometers in arbitrary units

actually is. The calibration procedure obviously depends strongly on the technical realization of the particular knowledge transferometer under assessment. Thus detailing the calibration procedures of knowledge transferometry is beyond the scope of this paper.

## 4.3 Validation of the Knowledge Transferometer

Often, validation has been reduced to a comparison of measurements by different systems in oder to find out if these mea-

surements agree within their error margin. The typical tool for this purpose is $\chi^2$ statistics (Pearson 1900; for application in geosciences, see, e.g., Rodgers 2000). If the probability that the discrepancy between the two sets of measurements is caused by a realization of the assumed combined measurement error distribution of the reference measurement and the measurement to be validated distribution is larger than 5%, then the differences are regarded as insignificant and the measurement is considered as validated.

Obviously the reference measurement used for validation has to be validated itself. If the quantity to be measured is defined in a theoretical manner, the validation problem leads either to an infinite regress or a logical circle. That means, that we either need an infinite chain of validation processes, each involving another measurement system, or at one time one of the measurement systems to be validated is used as reference instrument. In contrast, we are fine off since we have defined knowledge transfer in an operational manner, and the validation chain has to be extended only down to the final element of the chain, *viz.*, down to

the knowledge transferometer used for the definition of the unit Kant is reached.

## 4.4 Inversion of the Problem

Inspired by related literature (Adams, 1979) one might consider to attempt an inverse solution of the problem of the measurement of knowledge transfer. For this purpose it suggests itself to just assign a value to the knowledge transfer of the project under assessment. This value could be, e.g., fourty-two (in units of milliKant). The inversion part consists in the fact that some

meaning has to be assigned to this number but this side aspect can be left to the science managers who requested the measure-



ment of knowledge transfer. Some deeper thought reveals that this strategy is only adequate *cum grano salis* because it must be guaranteed that the measurement is not reported in Vogonian units.

## 5   Conclusion

Admittedly, the aim of this paper, to provide clarification with respect to the concepts and notions associated with knowledge
transfer, has not been reached. It is, however, hoped that at least some awareness has been created about what the open questions are and where the vagueness of this concept is hidden (or even obvious). Currently, the attempt to measure knowledge transfer seems to raise more questions than it answers. Clearly further research is needed within meta-scientific projects, which aim at answering the questions raised in this paper. Obviously these meta-scientific projects need evaluation themselves, including measurements of the transfer of the meta-knowledge gained (or meta-measurements of the knowledge gained, or even meta-
measurements of meta-knowledge gained). Unfortunately all three options seem to imply an infinite regress.

Another problem is that knowledge about knowledge, and also the transfer of knowledge about knowledge transfer imply some self-reference, and the first order logic applied to most parts of this paper might no longer suffice. This leads us into the vicinity of the work of Tarski, Gödel, Lindström and others. These considerations, however, are beyond the scope of this paper.

This finger exercise on the claviature of knowledge metrology has not solved any of the problems related to the measurement
of knowledge transfer, and it ends in an *aporia*, as did Theaitetos, Theodoros from Kyreme and Socrates (Plato, 369BC). In this fictive discussion the mathematicians Theaitetos and Theodoros, guided by Soctrates tried unsuccessfully to explicate the term 'knowledge'. Similarly as in the case of 'Theaitetos' it is hoped that the critical discussion in this paper is not regarded as a failure. Instead, the increased understanding where the problems are may help future proposals for solution to be more successful. At least, I hope to have transferred the knowledge on how incomplete our knowledge on knowledge transfer is. My
major fear, however, is that science managers, after seeing the mess they created by requiring the measurement of knowledge transfer, will just replace this term in their list of requirements by another impressive sounding but poorly defined and empty buzzword, just to keep scientists busy and to prevent them not only from fulfilling their primary scientific tasks but also from generating outreach towards the general public.

**Acknowledgment:** Thanks go to John Landis for his demonstration how to best structure ones ideas.





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

von Mises, R.: Kleines Lehrbuch des Positivismus – Einführung in die empiristische Wissenschaftsauffassung, Suhrkamp, Frankfurt am
5    Main, edited by Friedrich Stadler, 1990; original edition: Stockum und Zoon, Den Haag; english edition: "Positivism, a Study in Human Understanding", Cambridge, 1951., 1939.