# Peer review of "The Measurement of Knowledge Transfer"

_Geoscience Communication, 2018_

## Referee Comment (RC1) · D. Crookall (Referee) · 15 Jun 2018

Brief review by David Crookall, crookall.consulting@gmail.com, of The Measurement of Knowledge Transfer, by Thomas von Clarmann Geosci. Commun. Discuss., https://doi.org/10.5194/gc-2018-8 Manuscript under review for journal Geosci. Commun. ++++ Note to Thomas and to the Editor: I am a novice in epistemology and the philosophy of knowledge. At the outset, my impression, from having read a goodly portion of this article in detail and the rest by skimming, is that the article does not belong in Geoscience Communication (GC). Your ms does not even contain the word 'communication'!, and uses the term 'geo-science' only twice, and spelt in two different ways.

In my view, your article belongs in a journal that focusses on of the above two mentioned areas (epistemology and the philosophy of knowledge). However, a few random

notes may help you, Thomas, to develop your ms.

Objectives. At the outset, you need some clear objectives, such as: "... This article has three main objectives: a. To clarify concepts in, and to provide a framework for, knowledge and in the measurement of its transfer; b. To demonstrate how greater clarity and the framework may, on a practical, user level, serve to better communicate the complexity of meteorology and climate science; c. To provide a case study of a project in which the framework was deployed with encouraging results, in other words, how such insight can help further the work of geoscientists. ..."

It is, in my view, important to demonstrate the relevance of your work to down-to-earth communication in geoscience.

You should probably provide a lit rev (a) of the various concepts in general (philo of knowledge, epistemology, ethnomethodology*, phenomenology, etc), and (b) of their evolution and use in the geosciences. * Ethnomethodology is a powerful tool for analysing people's methods in sense making and social generation and legitimation of knowledge.

You could broaden out the lit rev to include ideas such as: • Data: Facts, a description of the World • Information: Captured Data and Knowledge • Knowledge: Our personal map/model of the World • Know-how: Skills at various levels; know-that • Intelligence & emotional intelligence • Making sense • Relating knowledge to action and to mind, as well as psychological and social process (eg, metaphors, interpretation, representation, simulation, etc).

You may find the Berger & Luckmann 'The social construction of reality' a deep source of ideas and clarity.

Finally, before you send your next draft, it is important to tighten up the English – grammar, phrasing, etc.

---

## Author Comment (AC1) · 19 Jun 2018

I have decided to withdraw this manuscript because I have become aware that a number of studies have recently been published that make this paper obsolete. Since I wrote this manuscript, a lot of work has been done which aims at clarification of the title term "measurement of knowledge transfer". My criticism of the vagueness of the title term seems to be no longer appropriate and I do not want researchers who work towards its clarification to feel criticized.